# Improving the Gating Mechanism of Recurrent Neural Networks

## Abstract

Gating mechanisms are widely used in neural network models, where they allow gradients to backpropagate more easily through depth or time. However, their saturation property introduces problems of its own. For example, in recurrent models these gates need to have outputs near 1 to propagate information over long time-delays, which requires them to operate in their saturation regime and hinders gradient-based learning of the gate mechanism. We address this problem by deriving two synergistic modifications to the standard gating mechanism that are easy to implement, introduce no additional hyperparameters, and improve learnability of the gates when they are close to saturation. We show how these changes are related to and improve on alternative recently proposed gating mechanisms such as chrono-initialization and Ordered Neurons. Empirically, our simple gating mechanisms robustly improve the performance of recurrent models on a range of applications, including synthetic memorization tasks, sequential image classification, language modeling, and reinforcement learning, particularly when long-term dependencies are involved.

## 1 Introduction

Recurrent neural networks (RNNs) have become a standard machine learning tool for learning from sequential data. However, RNNs are prone to the vanishing gradient problem, which occurs when the gradients of the recurrent weights become vanishingly small as they get backpropagated through time (Hochreiter et al., 2001). A common approach to alleviate the vanishing gradient problem is to use gating mechanisms, leading to models such as the long short term memory (Hochreiter & Schmidhuber, 1997, LSTM) and gated recurrent units (Chung et al., 2014, GRUs). These gated RNNs have been very successful in several different application areas such as in reinforcement learning (Kapturowski et al., 2018; Espeholt et al., 2018) and natural language processing (Bahdanau et al., 2014; Kočiský et al., 2018).

At every time step, gated recurrent models form a weighted combination of the history summarized by the previous state, and (a function of) the incoming inputs, to create the next state. The values of the gates, which are the coefficients of the combination, control the length of temporal dependencies that can be addressed. This weighted update can be seen as an additive or residual connection on the recurrent state, which helps signals propagate through time without vanishing. However, the gates themselves are prone to a saturating property which can also hamper gradient-based learning. This is particularly troublesome for RNNs, where carrying information for very long time delays requires gates to be very close to their saturated states.

We formulate and address two particular problems that arise with the standard gating mechanism of recurrent models. First, typical initialization of the gates is relatively concentrated. This restricts the range of timescales the model can address, as the timescale of a particular unit is dictated by its gates. Our first proposal, which we call *uniform gate initialization* (Section 2.2), addresses this by directly initializing the activations of these gates from a distribution that captures a wider spread of dependency lengths.

Second, learning when gates are in their saturation regime is difficult because of vanishing gradients through the gates. We derive a modification that uses an auxiliary *refine gate* to modulate a main gate, which allows it to have a wider range of activations without gradients vanishing as quickly.

Combining these two independent modifications yields our main proposal, which we call the **UR-gating** mechanism. These changes can be applied to any gate (i.e. bounded parametrized function) and have minimal to no overhead in terms of speed, memory, code complexity, and (hyper-)parameters.

We apply them to the forget gate of recurrent models, and evaluate on many benchmarks including synthetic long-term dependency tasks, sequential pixel-level image classification, language modeling, program execution, and reinforcement learning. Finally, we connect our methods to other proposed gating modifications, introduce a framework that allows each component to be replaced with similar ones, and perform theoretical analysis and extensive ablations of our method. Empirically, the UR- gating mechanism robustly improves on the standard forget and input gates of gated recurrent models. When applied to the LSTM, these simple modifications solve synthetic memory tasks that are pathologically difficult for the standard LSTM, achieve state-of-the-art results on sequential MNIST and CIFAR-10, and show consistent improvements in language modeling on the WikiText-103 dataset (Merity et al., 2016) and reinforcement learning tasks (Hung et al., 2018).

## 2 GATED RECURRENT NEURAL NETWORKS

Broadly speaking, RNNs are used to sweep over a sequence of input data $x_t$ to produce a sequence of recurrent states $h_t \in \mathbb{R}^d$ summarizing information seen so far. At a high level, an RNN is just a parametrized function in which each sequential application of the network computes a state update $u : (x_t, h_{t-1}) \mapsto h_t$. Gating mechanisms were introduced to address the vanishing gradient problem (Bengio et al., 1994; Hochreiter et al., 2001), and have proven crucial to the success of RNNs. This mechanism essentially smooths out the update using the following equation,

$$h_t = f_t(x_t, h_{t-1}) \circ h_{t-1} + i_t(x_t, h_{t-1}) \circ u(x_t, h_{t-1}), \tag{1}$$

where the *forget gate* $f_t$ and *input gate* $i_t$ are $[0,1]^d$-valued functions that control how fast information is forgotten or allowed into the memory state. When the gates are tied, i.e. $f_t + i_t = 1$ as in GRUs, they behave as a low-pass filter, deciding the time-scale on which the unit will respond (Tallec & Ollivier, 2018). For example, large forget gate activations close to $f_t = 1$ are necessary for recurrent models to address long-term dependencies.[1]

We will introduce our improvements to the gating mechanism primarily in the context of the LSTM, which is the most popular recurrent model. However, these techniques can be used in any model that makes similar use of gates.

A typical LSTM (equations (2)-(7)) is an RNN whose state is represented by a tuple $(h_t, c_t)$ consisting of a "hidden" state and "cell" state. The basic gate equation (1) is used to create the next cell state $c_t$ (5). Note that the gate and update activations are a function of the previous hidden state $h_{t-1}$ instead of $c_{t-1}$. Here, $\mathcal{L}_\star$ stands for a parameterized linear function of its inputs with bias $b_\star$, e.g.

$$f_t = \sigma(\mathcal{L}_f(x_t, h_{t-1})) \tag{2}$$
$$i_t = \sigma(\mathcal{L}_i(x_t, h_{t-1})) \tag{3}$$
$$u_t = \tanh(\mathcal{L}_u(x_t, h_{t-1})) \tag{4}$$
$$c_t = f_t \circ c_{t-1} + i_t \circ u_t \tag{5}$$
$$o_t = \sigma(\mathcal{L}_o(x_t, h_{t-1})) \tag{6}$$
$$h_t = o_t \tanh(c_t) \tag{7}$$

$$\mathcal{L}_f(x_t, h_{t-1}) = W_{fx} x_t + W_{fh} h_{t-1} + b_f, \tag{8}$$

and $\sigma(\cdot)$ refers to the standard sigmoid activation function which we will assume is used for defining $[0,1]$-valued activations in the rest of this paper. The gates of the LSTM were initially motivated as a binary mechanism, switching on or off to allow information and gradients to pass through. However, in reality this fails to happen due to a combination of initialization and saturation. This can be problematic, such as when very long dependencies are present.

### 2.1 THE UR-LSTM

We present two solutions which work in tandem to address the previously described issues. The first ensures a diverse range of gate values at the start of training by sampling the gate's biases so that the activations will be approximately uniformly distributed at initialization. We call this Uniform Gate Initialization (UGI). The second allows better gradient flow by reparameterizing the gate using an auxiliary "refine" gate.

As our main application is for recurrent models, we present the full UR-LSTM model in equations (9)-(15). However, we note that

$$b_f \sim \sigma^{-1}(\mathcal{U}[0,1]) \tag{9}$$
$$f_t = \sigma(\mathcal{L}_f(x_t, h_{t-1})) \tag{10}$$
$$r_t = \sigma(\mathcal{L}_r(x_t, h_{t-1})) \tag{11}$$
$$g_t = r_t \cdot (1 - (1 - f_t)^2)$$
$$\qquad + (1 - r_t) \cdot f_t^2 \tag{12}$$
$$c_t = g_t c_{t-1} + (1 - g_t) u_t \tag{13}$$
$$o_t = \sigma(\mathcal{L}_o(x_t, h_{t-1})) \tag{14}$$
$$h_t = o_t \tanh(c_t) \tag{15}$$

---

[1]In this work, we use "gate" to alternatively refer to a $[0,1]$-valued function or the value ("activation") of that function.

these methods can be used to modify any gate (or more generally, bounded function) in any model. In this context the UR-LSTM is simply defined by applying UGI and a refine gate $r$ on the original forget gate $f$ to create an effective forget gate $g$ (equation (12)). This effective gate is then used in the cell state update (13). Empirically, these small modifications to an LSTM are enough to allow it to achieve nearly binary activations and solve difficult memory problems (Figure 4). In the rest of Section 2, we provide theoretical justifications for UGI and refine gates.

## 2.2 UNIFORM GATE INITIALIZATION

Standard initialization schemes for the gates can prevent the learning of long-term temporal correlations (Tallec & Ollivier, 2018). For example, supposing that a unit in the cell state has constant forget gate value $f_t$, then the contribution of an input $x_t$ in $k$ time steps will decay by $(f_t)^k$. This gives the unit an effective *decay period* or *characteristic timescale* of $O(\frac{1}{1-f_t})$.[2] Standard initialization of linear layers $\mathcal{L}$ sets the bias term to $0$, which causes the forget gate values (2) to concentrate around $1/2$. A common trick of setting the forget gate bias to $b_f = 1.0$ (Jozefowicz et al., 2015) does increase the value of the decay period to $\frac{1}{1-\sigma(1.0)} \approx 3.7$. However, this is still relatively small, and moreover fixed, hindering the model from easily learning dependencies at varying timescales.

We instead propose to directly control the distribution of forget gates, and hence the corresponding distribution of decay periods. In particular, we propose to simply initialize the value of the forget gate activations $f_t$ according to a uniform distribution $\mathcal{U}(0,1)$, as described in Section 2.1. An important difference between UGI and standard or other (e.g. Tallec & Ollivier, 2018) initializations is that negative forget biases are allowed. The effect of UGI is that all timescales are covered, from units with very high forget activations remembering information (nearly) indefinitely, to those with low activations focusing solely on the incoming input. Additionally, it introduces no additional parameters; it even can have less hyperparameters than the standard gate initialization, which sometimes tunes the forget bias $b_f$. Appendix B.2 and B.3 further discuss the theoretical effects of UGI on timescales.

## 2.3 THE REFINE GATE

Given a gate $f = \sigma(\mathcal{L}_f(x)) \in [0,1]$, the refine gate is an independent gate $r = \sigma(\mathcal{L}_r(x))$, and modulates $f$ to produce a value $g \in [0,1]$ that will be used in place of $f$ downstream. It is motivated by considering how to modify the output of a gate $f$ in a way that promotes gradient-based learning, derived below.

**An additive modification**    The root of the saturation problem is that the gradient $\nabla f$ of a gate, which can be written solely as a function of the activation value as $f(1-f)$, decays rapidly as $f$ approaches $0$ or $1$. Thus when the activation $f$ is past a certain upper or lower threshold, learning effectively stops. This problem cannot be fully addressed only by modifying the input to the sigmoid, as in UGI and other techniques, as the gradient will still vanish by backpropagating through the activation function.

Therefore to better control activations near the saturating regime, instead of changing the input to the sigmoid in $f = \sigma(\mathcal{L}(x))$, we consider modifying the output. In particular, we consider adjusting $f$ with an input-dependent update $\phi(f,x)$ for some function $\phi$, to create an effective gate $g = f + \phi(f,x)$ that will be used in place of $f$ downstream such as in the main state update (1). This sort of additive ("residual") connection is a common technique to increase gradient flow, and indeed was the motivation of the LSTM additive gated update (1) itself (Hochreiter & Schmidhuber, 1997).

**Choosing the adjustment function**    Although many choices seem plausible for selecting the additive update $\phi$, we reason backwards from necessary properties of the effective activation $g$ to deduce a principled function $\phi$. The refine gate will appear as a result.

First, note that $f_t$ might need to be increased or decreased, regardless of what its value is. For example, given a large activation $f_t$ near saturation, it may need to be even higher to address long-term dependencies in recurrent models; alternatively, if it is too high by initialization or needs to unlearn previous behavior, it may need to decrease. Therefore, the additive update to $f$ should create an effective activation $g_t$ in the range $f_t \pm \alpha$ for some $\alpha$. Note that the allowed adjustment range $\alpha = \alpha(f_t)$ needs to be a function of $f$ in order to keep $g \in [0,1]$.

---

[2]This corresponds to the number of timesteps it takes to decay by $1/e$.

In particular, the additive adjustment range $\alpha(f)$ should satisfy the following natural properties:

*Validity*: $\alpha(f) \leq \min(f, 1-f)$, to ensure $g \in f \pm \alpha(f) \in [0,1]$.
*Symmetry*: Since 0 and 1 are completely symmetrical in the gating framework, $\alpha(f) = \alpha(1-f)$.
*Differentiability*: $\alpha(f)$ will be used in backpropagation, requiring $\alpha \in C^1(\mathbb{R})$.

Figure 2a illustrates the general appearance of $\alpha(f)$ based on these properties. In particular, Validity implies that that its derivative satisfies $\alpha'(0) \leq 1$ and $\alpha'(1) \geq -1$, Symmetry implies $\alpha'(f) = -\alpha'(1-f)$, and Differentiability implies $\alpha'$ is continuous. The simplest such function satisfying these is the linear $\alpha'(f) = 1 - 2f$, yielding $\alpha(f) = f - f^2 = f(1-f)$.

Given such a $\alpha(f)$, recall that the goal is to produce an effective activation $g = f + \phi(f, x)$ such that $g \in f \pm \alpha(f)$ (Figure 2b). Our final observation is that the simplest such function $\phi$ satisfying this is $\phi(f, x) = \alpha(f)\psi(f, x)$ for some $\psi(\cdot) \in [-1, 1]$. Using the standard method for defining $[-1, 1]$-valued functions via a $\tanh$ non-linearity leads to $\phi(f, x) = \alpha(f)(2r - 1)$ for another gate $r = \sigma(\mathcal{L}(x))$.

The full update is given in Equation (16),

$$g = f + \alpha(f)(2r-1) = f + f(1-f)(2r-1) = (1-r) \cdot f^2 + r \cdot (1-(1-f)^2) \quad (16)$$

Equation (16) has the elegant interpretation that the gate $r$ linearly interpolates between the lower band $f - \alpha(f) = f^2$ and the symmetric upper band $f + \alpha(f) = 1 - (1-f)^2$ (Figure 2b). In other words, the original gate $f$ is the coarse-grained determinant of the effective gate $g$, while the gate $r$ "refines" it. This allows the effective gate $g$ to reach much higher and lower activations than the constituent gates $f$ and $r$ (Figure 2c), bypassing the saturating gradient problem. For example, this allows the effective forget gate to reach $g = 0.99$ when the forget gate is only $f = 0.9$.

## 2.4 REFINING RECURRENT MODELS

Formally, the full mechanism of the refine gate as applied to gated recurrent models is defined in equations (11)-(13). Note that it is an isolated change where the forget gate (10) is modified before applying the the standard update (1). Figure 1 illustrates the refine gate in an LSTM cell. Figure 2 illustrates how the refine gate $r_t$ is defined and how it changes the effective gate $f_t$ to produce an effective gate $g_t$.

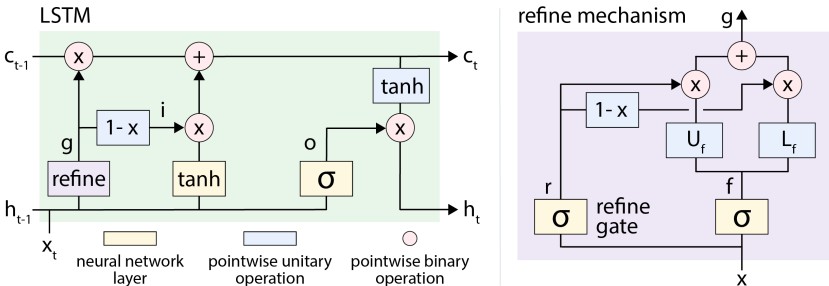

Figure 1: **LSTM with refine gate** The refine gate $r_t$ modifies another gate, such as the forget gate $f_t$ for recurrent models. It interpolates between upperbound $U_{f_t}$ and lowerbound $L_{f_t}$ functions of the forget gate. The resulting effective forget gate $g_t$ is then used in place of $f_t$ in the state update (5).

Finally, to simplify comparisons and ensure that we always use the same number of parameters as the standard gates, when using the refine gate we tie the input gate to the effective forget gate, $i_t = 1 - g_t$. However, we emphasize that these techniques are extremely simple and broad, and can be applied to any gate (or more broadly, any bounded function) to improve initialization distribution and help optimization. For example, our methods can be combined in different ways in recurrent models, e.g. an independent input gate can be modified with its own refine gate. Alternatively, the refine gate can also be initialized uniformly, which we do in our experiments whenever both UGI and refine gates are used.

## 2.5 RELATED GATING MECHANISMS

We highlight a few recent works that also propose small gate changes to address problems of long-term or variable-length dependencies. Like ours, they can be applied to any gated update equation.

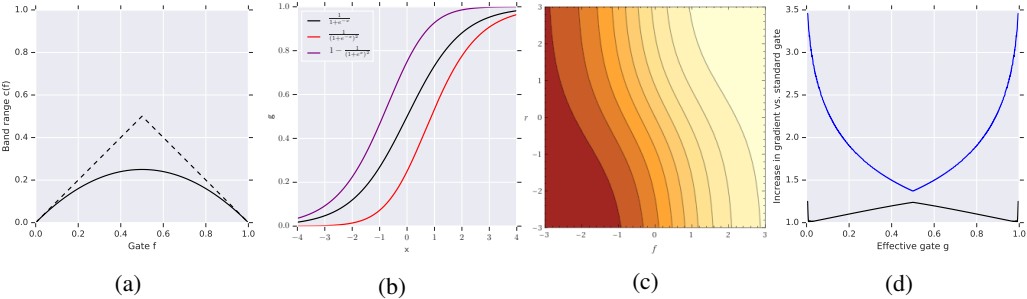

Figure 2: **Refine gate** in action: (a) [Solid] A function $\alpha(f_t)$ satisfying natural properties is chosen to define a band within which the forget gate is refined. (b) The forget gate $f_t(x)$ is conventionally defined with the sigmoid function (black). The refine gate interpolates around the original gate $f_t$ to yield an effective gate $g_t$ within the upper and lower curves, $g_t \in f_t \pm \alpha(f_t)$. (c) Contours of the effective gate $g_t$ as a function of the forget and refine gates $f_t, r_t$. High effective activations can be achieved with more modest $f_t, r_t$ values. (d) The gradient $\nabla g_t$ as a function of effective gate activation $g_t$. [Black, blue]: Lower and upper bounds on the ratio of the gradient when using a refine gate vs. without.

Tallec & Ollivier (2018) suggest an initialization strategy to capture long-term dependencies on the order of $T_{max}$, by sampling the gate biases from $b_f \sim \log \mathcal{U}(1, T_{max}-1)$. Although similar to UGI in definition, *chrono initialization* (CI) has key differences in the timescales captured, for example by using an explicit timescale parameter and having no negative biases. Due to its relation to UGI, we provide a more detailed comparison in Appendix B.3. As mentioned in Section 2.3, techniques such as these that only modify the input to a sigmoid gate do not fully address the saturation problem.

The Ordered Neuron LSTM introduced by Shen et al. (2018) aims to induce an ordering over the units in the hidden states such that "higher-level" neurons retain information for longer and capture higher-level information. We highlight this work due to its recent success in NLP, and also because its novelties can be factored into introducing two mechanisms which only affect the forget and input gates, namely (i) the $\mathrm{cumax} := \mathrm{cumsum} \circ \mathrm{softmax}$ activation function which creates a monotonically increasing vector in $[0,1]$, and (ii) a pair of "master gates" which are ordered by $\mathrm{cumax}$ and fine-tuned with another pair of gates.

In fact, we observe that these are related to our techniques in that one controls the distribution of a gate activation, and the other is an auxiliary gate with modulating behavior. Despite its important novelties, we find that the ON-LSTM has drawbacks including speed/stability issues and theoretical flaws in the scaling of its gates. We provide the formal definition and detailed analysis of the ON-LSTM in Appendix B.4. In particular we flesh out a deeper relation between the master and refine gates and show how they can be interchanged for each other.

We include a more thorough overview of other related works on RNNs in Appendix B.1. These methods are largely orthogonal to the isolated gate changes considered here and are not analyzed. We note that an important drawback common to all other approaches is the introduction of substantial hyperparameters in the form of constants, training protocol, and significant architectural changes. For example, even for chrono initialization, one of the less intrusive proposals, we experimentally find it to be particularly sensitive to the hyperparameter $T_{max}$ (Section 3).

### 2.6 GATE ABLATIONS

Our insights about previous work with related gate components allow us to perform extensive ablations of our contributions. We observe two independent axes of variation, namely, activation function/initialization ($\mathrm{cumax}$, constant bias sigmoid, CI, UGI) and auxiliary modulating gates (master, refine), where different components can be replaced with each other. Therefore we propose several other gate combinations to isolate the effects of different gating mechanisms. We summarize a few ablations here; precise details are given in Appendix B.5. **O-: Ordered gates**. A natural simplification of the main idea of ON-LSTM, while keeping the hierarchical bias on the forget activations, is to simply drop the auxiliary master gates and define $f_t, i_t$ (2)-(3) using the $\mathrm{cumax}$ activation function. **UM-: UGI master gates**. This variant of the ON-LSTM's gates ablates the $\mathrm{cumax}$ operation on the master gates,

replacing it with a sigmoid activation and UGI which maintains the same initial distribution on the activation values. **OR-: Refine instead of master**. A final variant in between the UR- gates and the ON-LSTM's gates combines cumax with refine gates. In this formulation, as in UR- gates, the refine gate modifies the forget gate and the input gate is tied to the effective forget gate. The forget gate is ordered using cumax.

Table 1 summarizes the gating modifications we consider and their naming conventions. Note that we also denote the ON-LSTM method as "OM-LSTM" (M for master) for mnemonic ease. Finally, we remark that all methods here are controlled with the same number of parameters as the standard LSTM, aside from the OM-LSTM and UM-LSTM which use an additional $\frac{1}{2C}$-fraction parameters where $C$ is the downsize factor on the master gates (Appendix B.4).

Table 1: Summary of gating mechanisms considered in this work as applied to the main forget/input gates of recurrent models. To preserve parameters, refine gate methods use a tied input gate, and master gate methods use a downsize factor $C > 1$. (Left) Existing approaches and our main method. (right) Ablations of our gates with different components.

| Name | Gate Mechanism | | |
|------|----------------|------|--------------------------------------------------|
| | | U- | Uniform gate initialization, no auxiliary gate |
| - | Standard gate initialization (1) | R- | Refine gate with standard gate initialization |
| C- | Chrono initialization, no auxiliary gate | O- | cumax activation on forget/input gates |
| OM- | Ordered main gates, auxiliary master gates | UM- | UGI main gates, auxiliary master gates |
| **UR-** | UGI main gate, auxiliary refine gate | OR- | Ordered main gate, auxiliary refine gate |

## 3 EXPERIMENTS

We first perform full ablations of the gating variants (Section 2.6) on benchmark synthetic memorization and pixel-by-pixel image classification tasks. We then evaluate our main method on important applications for recurrent models including language modeling and reinforcement learning, comparing against baseline methods where appropriate.

The vanilla LSTM uses forget bias 1.0 (Section 2.2). When chrono-initialization is used and not explicitly tuned, we set $T_{max}$ to be proportional to the hidden size. This heuristic uses the intuition that if dependencies of length $T$ exist, then so should dependencies of all lengths $\leq T$. Moreover, the amount of information that can be remembered is proportional to the number of hidden units.

All of our benchmarks have prior work with recurrent baselines, from which we used the same models, protocol, and hyperparameters whenever possible, changing only the gating mechanism. Since our simple gate changes are compatible with other recurrent cores, we evaluate them in tandem with recurrent models such as the GRU, Reconstructive Memory Agent (RMA; Hung et al., 2018), and Relational Memory Core (RMC; Santoro et al., 2018) whenever they were used on these tasks. Full protocols and details for all experiments are given in Appendix D.

### 3.1 SYNTHETIC TASKS

Our first set of experiments is on synthetic memory tasks (Hochreiter & Schmidhuber, 1997; Arjovsky et al., 2016) that are known to be hard for standard LSTMs to solve.

**Copy task.** The input is a sequence of $N + 20$ digits where the first 10 tokens $(a_0, a_1, ..., a_9)$ are randomly chosen from $\{1, ..., 8\}$, the middle N tokens are set to 0, and the last ten tokens are 9. The goal of the recurrent model is to output $(a_0, ..., a_9)$ in order on the last 10 time steps, whenever the cue token 9 is presented. We trained our models using cross-entropy with baseline loss $\log(8)$ (Appendix D.1).

**Adding task.** The input consists of two sequences: 1. $N$ numbers $(a_0, ..., a_{N-1})$ sampled independently from $\mathcal{U}[0, 1]$ 2. an index $i_0 \in [0, N/2)$ and $i_1 \in [N/2, N)$, together encoded as a two-hot sequence. The target output is $a_{i_0} + a_{i_1}$ and models are evaluated by the mean squared error with baseline loss $1/6$.

Figure 3 shows the loss of various methods on the Copy and Adding tasks. The only gate combinations capable of solving Copy completely are **OR-**, **UR-**, **O-**, and **C-LSTM**. This confirms the mechanism of their gates: these are the only methods capable of producing high enough forget gate values either through the cumax non-linearity, the refine gate, or extremely high forget biases. The U-LSTM is the only other method able to make progress, but converges slower as it suffers from gate saturation

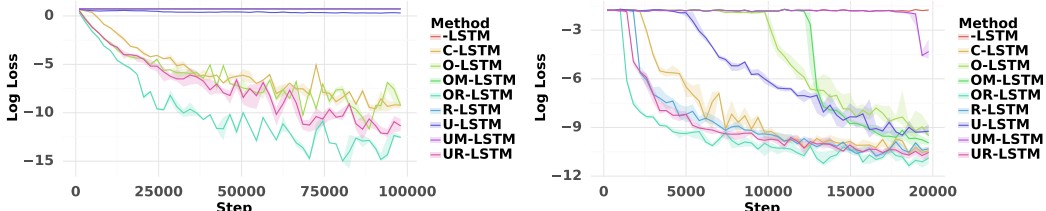

Figure 3: (Left) Copy task length 500 (Right) Adding task length 2000. Every method besides the LSTM solves the Adding task. The only methods capable of solving copy are OR-,UR-,O-, and C-LSTM models, with all other models aside from U-LSTM stuck at baseline. Refine gates are fastest.

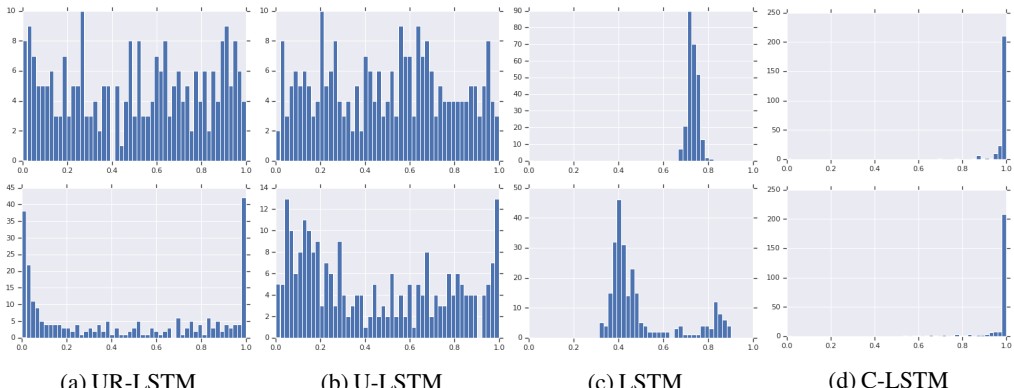

(a) UR-LSTM      (b) U-LSTM      (c) LSTM      (d) C-LSTM

Figure 4: Histograms of forget gate $f_t$ activations (averaged over time and batch) before (Top) and after (Bottom) training on Copy (y-axis independently scaled). C-LSTM initializes with extremal activations which barely change during training. Standard LSTM initialization cannot learn large enough $f_t$ and makes no progress on the task. U-LSTM makes progress by encouraging a range of forget gate values, but this distribution does not change significantly during training due to saturation. UR-LSTM starts with the same distribution, but is able to learn extremal gate values. Complementary to here when learning large activations is necessary, Appendix E.1 shows a reverse task where the UR-LSTM is able to un-learn from a saturated regime.

without the refine gate. The vanilla LSTM makes no progress. The OM-LSTM and UM-LSTM also get stuck at the baseline loss, despite the OM-LSTM's cumax activation, which we hypothesize is due to the suboptimal magnitudes of the gates at initialization (Appendix B.4). On the Adding task, every method besides the basic LSTM is able to eventually solve it, with all refine gate variants fastest.

Figure 4 shows the distributions of forget gate activations of sigmoid-activation methods, before and after training on the Copy task. It shows that activations near $1.0$ are important for a model's ability to make progress or solve this task, and that adding the refine gate makes this significantly easier.

## 3.2 PIXEL-BY-PIXEL IMAGE CLASSIFICATION

These tasks involve feeding a recurrent model the pixels of an image in a scanline order before producing a classification label. We test on the sequential MNIST (sMNIST), permuted MNIST (pMNIST) (Le et al., 2015), and sequential CIFAR-10 (sCIFAR) tasks. Each LSTM method was ran with a learning rate sweep with 3 seeds each. The best validation score found over any run is reported in the first two rows of Table 2.[3] We find in general that all methods are able to improve over the vanilla LSTM. However, the differences become even more pronounced when stability is considered. Although Table 2 reports the best validation accuracies found on any run, we found that many methods were quite unstable. Asterisks are marked next to a score denoting how many of the 3 seeds diverged, for the learning rate that score was found at.

---

[3] sMNIST is not included here as it is too easy, making it difficult to draw conclusions.

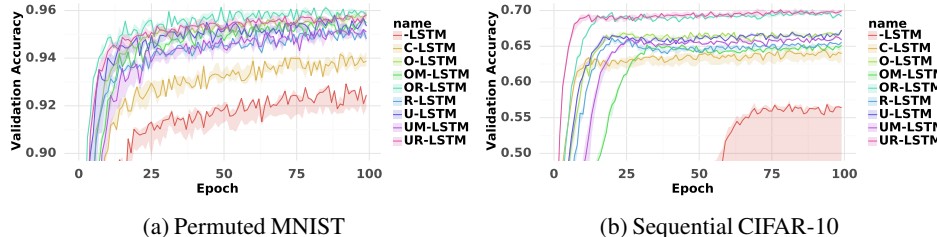

(a) Permuted MNIST            (b) Sequential CIFAR-10

Figure 5: Learning curves with deviations on pixel image classification, at the best stable learning rate.

Conversely, Figure 5 shows the accuracy curves of each method at their best *stable* learning rate. The basic LSTM is noticeably worse than all of the others. This suggests that any of the gate modifications, whether better initialization, cumax non-linearity, or master or refine gates, are better than standard gates especially when long-term dependencies are present. Additionally, the uniform gate initialization methods are generally better than the ordered and chrono initialization, and the refine gate performs better than the master gate. We additionally consider applying other techniques developed for recurrent models that are independent of the gating mechanism. Table 2 also reports scores when the same gating mechanisms are applied to the GRU model instead of the LSTM, where similar trends hold across the gating variants. In particular, UR-GRU is the only method that is able to stably attain good performance. As another example, the addition of a generic regularization technique—we chose Zoneout (Krueger et al., 2016) with default hyperparameters ($z_c = 0.5$, $z_h = 0.05$)—continued improving the UR-LSTM/GRU, outperforming even non-recurrent models on sequential MNIST and CIFAR-10. Table 3 compares the test accuracy of our main model against other models.

Table 2: Validation accuracies on pixel image classification. Asterisks denote divergent runs at the learning rate the best validation score was found at.

| Gating Method | - | C- | O- | U- | R- | OM- | OR- | UM- | UR- |
|---|---|---|---|---|---|---|---|---|---|
| pMNIST | 94.77** | 94.69 | 96.17 | 96.05 | 95.84* | 95.98 | 96.40 | 95.50 | 96.43 |
| sCIFAR | 63.24** | 65.60 | 67.78 | 67.63 | 71.85* | 67.73* | 70.41 | 67.29* | 71.05 |
| sCIFAR (GRU) | 71.30* | 64.61 | 69.81** | 70.10 | 70.74* | 70.20* | 71.40** | 69.17* | 71.04 |

Table 3: Test acc. on pixel-by-pixel image classification benchmarks. Top: Recurrent baselines and variants. Middle: Non-recurrent sequence models with global receptive field. Bottom: Our methods.

| Model | sMNIST | pMNIST | sCIFAR |
|---|---|---|---|
| LSTM (ours) | 98.9 | 95.11 | 63.01 |
| Dilated GRU (Chang et al., 2017) | 99.0 | 94.6 | - |
| IndRNN (Li et al., 2018a) | 99.0 | 96.0 | - |
| r-LSTM (2-Layer with Auxiliary Loss) (Trinh et al., 2018) | 98.4 | 95.2 | 72.2 |
| Transformer (Trinh et al., 2018) | 98.9 | 97.9 | 62.2 |
| Temporal convolution network (Bai et al., 2018a) | 99.0 | 97.2 | |
| TrellisNet (Bai et al., 2018b) | 99.20 | **98.13** | 73.42 |
| UR-LSTM | **99.28** | 96.96 | 71.00 |
| UR-LSTM + Zoneout (Krueger et al., 2016) | 99.21 | 97.58 | 74.34 |
| UR-GRU + Zoneout | 99.27 | 96.51 | **74.4** |

From Sections 3.1 and 3.2, we draw a few conclusions about the comparative performance of different gate modifications. First, the refine gate is consistently better than comparable master gates. CI solves the synthetic memory tasks but is worse than any other variant outside of those. We find ordered (cumax) gates to be effective, but speed issues prevent us from using them in more complicated tasks. UR- gates are consistently among the best performing and most stable.

| Model | Valid. | Test |
|-------|--------|------|
| LSTM | 34.3 | 35.8 |
| C-LSTM | 35.0 | 36.4 |
| C-LSTM ($T_{max} = 8$) | 34.3 | 36.1 |
| C-LSTM ($T_{max} = 11$) | 34.6 | 35.8 |
| OM-LSTM | 34.0 | 34.7 |
| U-LSTM | 33.8 | 34.9 |
| **UR-LSTM** | **33.6** | **34.6** |

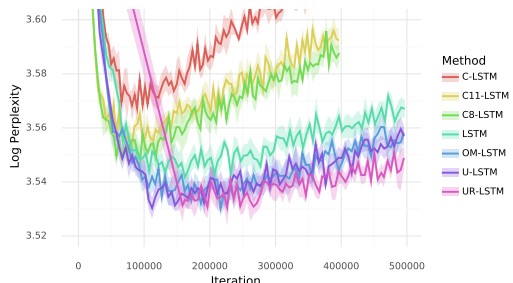

(a) Perplexities on the WikiText-103 dataset.

(b) Validation learning curves, illustrating training speed and generalization (i.e. overfitting) behavior.

### 3.3 LANGUAGE MODELING

We consider word-level language modeling on the WikiText-103 dataset, where (i) the dependency lengths are much shorter than in the synthetic tasks, (ii) language has an implicit hierarchical structure and timescales of varying lengths. We evaluate our gate modifications against the exact hyperparameters of a SOTA LSTM-based baseline (Rae et al., 2018) without additional tuning (Appendix D). Additionally, we compare against ON-LSTM, which was designed for this domain (Shen et al., 2018), and chrono initialization, which addresses dependencies of a particular timescale as opposed to timescale-agnostic UGI methods. In addition to our default hyperparameter-free initialization, we tested models with the chrono hyperparameter $T_{max}$ manually set to 8 and 11, values previously used for language modeling to mimic fixed biases of about 1.0 and 2.0 respectively (Tallec & Ollivier, 2018).

Table 6a shows Validation and Test set perplexities for various models. We find that the OM-LSTM, U-LSTM, and UR-LSTM improve over the standard LSTM with no additional tuning. However, although the OM-LSTM was designed to capture the hierarchical nature of language with the cumax activation, it does not perform better than the U-LSTM and UR-LSTM. The chrono initialization with our default initialization strategy is far too large. While manually tweaking the $T_{max}$ hyperparameter helps, it is still far from any UGI-based methods. We attribute these observations to the nature of language having dependencies on multiple widely-varying timescales, and that UGI is enough to capture these without resorting to strictly enforced hierarchies such as in OM-LSTM.

### 3.4 REINFORCEMENT LEARNING

In many partially observable reinforcement learning (RL) tasks, the agent can observe only part of the environment at a time and thus requires a memory model to summarize what it has seen previously. However, designing memory architectures for reinforcement learning problems has been a challenging task (Oh et al., 2016; Wayne et al., 2018). These are usually based on an LSTM core to summarize what an agent has seen into a state.

We investigated if changing the gates of these recurrent cores can improve the performance of RL agents, especially on difficult tasks involving memory and long-term credit assignment. We chose the Passive and Active Image Match tasks from Hung et al. (2018) using A3C agents (Mnih et al., 2016). In these tasks, agents are either initially shown a colored indicator (Passive) or must search for it (Active), before being teleported to a room in which they must press a switch with matching color to receive reward. In between these two phases is an intermediate phase where they can acquire *distractor* rewards, but the true objective reported is the final reward in the last phase. Episodes last $450 - 600$ steps, so these tasks require memorization and credit assignment across long sequences.

Hung et al. (2018) evaluated agents with different recurrent cores: the basic LSTM, the DNC (an LSTM with memory), and the RMA (which also uses an LSTM core). We modified each of these with our gates. Figure 7 shows the results of different models on the Passive Matching and Active Matching tasks without distractors. These are the most similar to the synthetic tasks (Sec. 3.1), and we found that those trends largely transferred to the RL setting even with several additional confounders present such as agents learning via RL algorithms, being required to learn relevant features from pixels rather than being given the relevant tokens, and being required to explore in the Active Match case.

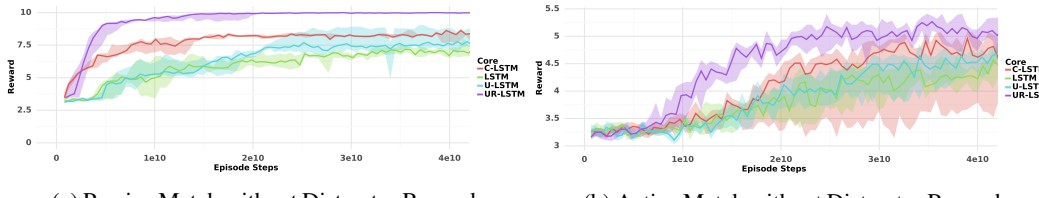

(a) Passive Match without Distractor Rewards    (b) Active Match without Distractor Rewards

Figure 7: We evaluated the image matching tasks from Hung et al. (2018), which test memorization and credit assignment, using an A3C agent (Mnih et al., 2016) with an LSTM policy core. We observe that general trends from the synthetic tasks (Section (3.1)) transfer to this reinforcement learning setting.

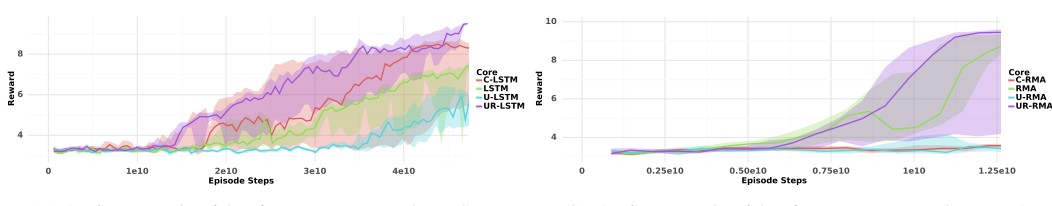

(a) Active Match with Distractor Rewards - LSTM    (b) Active Match with Distractor Rewards - RMA

Figure 8: The addition of distractor rewards changes the task and relative performance of different gating mechanisms. For both LSTM and RMA recurrent cores, the UR- gates still perform best.

We found that the UR- gates substantially improved the performance of the basic LSTM core on both Passive Match and Active Match tasks, with or without distractor rewards. On the difficult Active Match task, it was the only method to achieve better than random behavior. Figure 8 shows performance of LSTM and RMA cores on the harder Active Match task with distractors. Here the UR- gates again learn the fastest and reach the highest reward. In particular, although the RMA is a memory architecture with an explicit memory bank designed for long-term credit assignment, its performance was also improved.

### 3.5    ADDITIONAL RESULTS AND EXPERIMENTAL CONCLUSIONS

Appendix (E.1) shows an additional synthetic experiment investigating the effect of refine gates on saturation. Appendix (E.3) has results on a program execution task, which is interesting for having explicit long and variable-length dependencies and hierarchical structure. It additionally shows another very different gated recurrent model where the UR- gates provide consistent improvement.

Finally, we would like to comment on the longevity of the LSTM, which for example was frequently found to outperform newer competitors when tuned better (Melis et al., 2017). Although many improvements have been suggested over the years, none have been proven to be as robust as the LSTM across an enormously diverse range of sequence modeling tasks. By experimentally starting from well-tuned LSTM baselines, we believe our simple isolated gate modifications to actually be robust improvements. In Appendix B.3 and B.4, we offer a few conclusions for the practitioner about the other gate components considered based on our experimental experience.

## 4    DISCUSSION

In this work, we introduce, analyze, and evaluate several modifications to the ubiquitous gating mechanism that appears in recurrent neural networks. We describe theoretically-justified methods that improve on the standard gating method by alleviating problems with initialization and optimization. The mechanisms considered include changes on independent axes, namely initialization method and auxiliary gates, and we perform extensive ablations on our improvements with previously considered modifications. Our main gate model robustly improves on standard gates across many different tasks and recurrent cores, while requiring less tuning Finally, we emphasize that these improvements are completely independent of the large body of research on neural network architectures that use gates, and hope that these insights can be applied to improve machine learning models at large.

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

## A   PSEUDOCODE

We show how the gated update in a typical LSTM implementation can be easily replaced by UR- gates.

The following snippets show pseudocode for the the gated state updates for a vanilla LSTM model (top) and UR-LSTM (bottom).

```
forget_bias = 1.0 # hyperparameter
...
f, i, u, o = Linear(x, prev_hidden)
f_ = sigmoid(f + forget_bias)
i_ = sigmoid(i)
next_cell = f_ * prev_cell + i_ * tanh(u)
next_hidden = o * tanh(next_cell)
```

Listing 1: LSTM

```
# Initialization
u = np.random.uniform(low=1/hidden_size, high=1-1/hidden_size, hidden_size)
forget_bias = -np.log(1/u-1)
...
# Recurrent update
f, r, u, o = Linear(x, prev_hidden)
f_ = sigmoid(f + forget_bias)
r_ = sigmoid(r - forget_bias)
g = 2*r_*f_ + (1-2*r_)*f_**2
next_cell = g * prev_cell + (1-g) * tanh(u)
next_hidden = o * tanh(next_cell)
```

Listing 2: UR-LSTM

## B  FURTHER DISCUSSION ON RELATED METHODS

Section 2.5 briefly introduced chrono initialization (Tallec & Ollivier, 2018) and the ON-LSTM (Shen et al., 2018), closely related methods that modify the gating mechanism of LSTMs. We provide more detailed discussion on these in Sections B.3 and B.4 respectively. Section B.1 has a more thorough overview of related work on recurrent neural networks that address long-term dependencies or saturating gates.

### B.1  RELATED WORK

Several methods exist for addressing gate saturation or allowing more binary activations. Gulcehre et al. (2016) proposed to use piece-wise linear functions with noise in order to allow the gates to operate in saturated regimes. Li et al. (2018b) instead use the Gumbel trick (Maddison et al., 2016; Jang et al., 2016), a technique for learning discrete variables within a neural network, to train LSTM models with discrete gates. These stochastic approaches can suffer from issues such as gradient estimation bias, unstable training, and limited expressivity from discrete instead of continuous gates. Additionally they require more involved training protocols with an additional temperature hyperparameter that needs to be tuned explicitly.

Alternatively, gates can be removed entirely if strong constraints are imposed on other parts of the model. (Li et al., 2018a) use diagonal weight matrices and require stacked RNN layers to combine information between hidden units. A long line of work has investigated the use of identity or orthogonal initializations and constraints on the recurrent weights to control multiplicative gradients unrolled through time (Le et al., 2015; Arjovsky et al., 2016; Henaff et al., 2016). Chandar et al. (2019) proposed another RNN architecture using additive state updates and non-saturating activation functions instead of gates. However, although these gate-less techniques can be used to alleviate the vanishing gradient problem with RNNs, unbounded activation functions can cause less stable learning dynamics and exploding gradients.

As mentioned, a particular consequence of the inability of gates to approach extrema is that gated recurrent models struggle to capture very long dependencies. These problems have traditionally been addressed by introducing new components to the basic RNN setup. Some techniques include stacking layers in a hierarchy (Chung et al., 2016), adding skip connections and dilations (Koutnik et al., 2014; Chang et al., 2017), using an external memory (Graves et al., 2014; Weston et al., 2014; Wayne et al., 2018; Gulcehre et al., 2017), auxiliary semi-supervision (Trinh et al., 2018), and more. However, these approaches have not been widely adopted over the standard LSTM as they are often specialized for certain tasks, are not as robust, and introduce additional complexity. Recently the transformer model has been successful in many applications areas such as NLP (Radford et al., 2019; Dai et al., 2019). However, recurrent neural networks are still important and commonly used due their faster inference without the need to maintain the entire sequence in memory. We emphasize that the vast majority of proposed RNN changes are completely orthogonal to the simple gate improvements in this work, and we do not focus on them.

A few other recurrent cores that use the basic gated update (1) but use more sophisticated update functions $u$ include the GRU, Reconstructive Memory Agent (RMA; Hung et al., 2018), and Relational Memory Core (RMC; Santoro et al., 2018), which we consider in our experiments.

### B.2  EFFECT OF PROPOSED METHODS ON TIMESCALES

We briefly review the connection between our methods and the effective timescales that gated RNNs capture. Recall that Section 2.2 defines the *characteristic timescale* of a neuron with forget activation $f_t$ as $1/(1-f_t)$, which would be the number of timesteps it takes to decay that neuron by a constant.

The fundamental principle of gated RNNs is that the activations of the gates affects the timescales that the model can address; for example, forget gate activations near $1.0$ are necessary to capture long-term dependencies.

Thus, although our methods were defined in terms of activations $g_t$, it is illustrative to reason with their characteristic timescales $1/(1-g_t)$ instead, whence both UGI and refine gate also have clean interpretations.

First, UGI is equivalent to initializing the decay period from a particular heavy-tailed distribution, in contrast to standard initialization with a fixed decay period $(1-\sigma(b_f))^{-1}$.

**Proposition 1.** *UGI is equivalent to to sampling the decay period $D = 1/(1-f_t)$ from a distribution with density proportional to $\mathbb{P}(D=x) \propto \frac{d}{dx}(1-1/x) = x^{-2}$, i.e. a Pareto($\alpha=2$) distribution.*

On the other hand, for any forget gate activation $f_t$ with timescale $D = 1/(1-f_t)$, the refine gate fine-tunes it between $D = 1/(1-f_t^2) = 1/(1-f_t)(1+f_t)$ and $1/(1-f_t)^2$.

**Proposition 2.** *Given a forget gate activation with timescale $D$, the refine gate creates an effective forget gate with timescale in $(D/2, D^2)$.*

### B.3   Chrono Initialization

The chrono initialization

$$b_f \sim \log(\mathcal{U}([1, T_{max}-1])) \tag{17}$$
$$b_i = -b_f. \tag{18}$$

was the first to explicitly attempt to initialize the activation of gates across a distributional range. It was motivated by matching the gate activations to the desired timescales.

They also elucidate the benefits of tying the input and forget gates, leading to the simple trick (18) for approximating tying the gates at initialization, which we borrow for UGI. (We remark that perfect tied initialization can be accomplished by fully tying the linear maps $\mathcal{L}_f, \mathcal{L}_i$, but (18) is a good approximation.)

However, the main drawback of CI is that the initialization distribution is too heavily biased toward large terms. This leads to empirical consequences such as difficult tuning (due to most units starting in the saturation regime, requiring different learning rates) and high sensitivity to the hyperparameter $T_{max}$ that represents the maximum potential length of dependencies. For example, Tallec & Ollivier (2018) set this parameter according to a different protocol for every task, with values ranging from 8 to 2000. Our experiments used a hyperparameter-free method to initialize $T_{max}$ (Section 3), and we found that chrono initialization generally severely over-emphasizes long-term dependencies if $T_{max}$ is not carefully controlled.

A different workaround suggested by Tallec & Ollivier (2018) is to sample from $\mathbb{P}(T=k) \propto \frac{1}{k \log^2(k+1)}$ and setting $b_f = \log(T)$. Note that such an initialization would be almost equivalent to sampling the decay period from the distribution with density $\mathbb{P}(D=x) \propto (x \log^2 x)^{-1}$ (since the decay period is $(1-f)^{-1} = 1 + \exp(b_f)$). This parameter-free initialization is thus similar in spirit to the uniform gate initialization (Proposition 1), but from a much heavier-tailed distribution that emphasizes very long-term dependencies.

These interpretations suggest that it is plausible to define a family of Pareto-like distributions from which to draw the initial decay periods from, with this distribution treated as a hyperparameter. However, with no additional prior information on the task, we believe the uniform gate initialization to be the best candidate, as it 1. is a simple distribution with easy implementation, 2. has characteristic timescale distributed as an intermediate balance between the heavy-tailed chrono initialization and sharply decaying standard initialization, and 3. is similar to the ON-LSTM's cumax activation, in particular matching the initialization distribution of the cumax activation.

Table 4 summarizes the decay period distributions at initialization using different activations and initialization strategies.

In general, our experimental recommendation for CI is that it can be better than standard initialization or UGI when certain conditions are met (tasks with long dependencies and nearly fixed-length sequences as in Sections 3.1, 3.4) and/or when it can be explicitly tuned (both the hyperparameter $T_{max}$, as well as the learning rate to compensate for almost all units starting in saturation). Otherwise, we recommend UGI or standard initialization. We found no scenarios where it outperformed UR- gates.

### B.4   ON-LSTM

In this section we elaborate on the connection between the mechanism of Shen et al. (2018) and our methods. We define the full ON-LSTM and show how its gating mechanisms can be improved. For

Table 4: Distribution of the decay period $D = (1-f)^{-1}$ using different initialization strategies.

| Initialization method | Timescale distribution |
| --- | --- |
| Constant bias $b_f = b$ | $\mathbb{P}(D=x) \propto \mathbf{1}\{x = 1 + e^b\}$ |
| Chrono initialization (known timescale $T_{max}$) | $\mathbb{P}(D=x) \propto \mathbf{1}\{x \in [2, T_{max}]\}$ |
| Chrono initialization (unknown timescale) | $\mathbb{P}(D=x) \propto \frac{1}{x\log^2 x}$ |
| Uniform gate initialization | $\mathbb{P}(D=x) \propto \frac{1}{x^2}$ |
| cumax activation | $\mathbb{P}(D=x) \propto \frac{1}{x^2}$ |

example, there is a remarkable connection between its master gates and our refine gates – independently of the derivation of refine gates in Section 2.3, we show how a specific way of fixing the normalization of master gates becomes equivalent to a single refine gate.

First, we formally define the full ON-LSTM. The master gates are a cumax-activation gate

$$\tilde{f}_t = \mathrm{cumax}(\mathcal{L}_{\tilde{f}}(x_t, h_{t-1})) \tag{19}$$

$$\tilde{i}_t = 1 - \mathrm{cumax}(\mathcal{L}_{\tilde{i}}(x_t, h_{t-1})). \tag{20}$$

These combine with an independent pair of forget and input gates $f_t, i_t$, meant to control fine-grained behavior, to create an effective forget/input gate $\hat{f}_t, \hat{i}_t$ which are used to update the state (equation (1) or (5)).

$$\omega_t = \tilde{f}_t \circ \tilde{i}_t \tag{21}$$

$$\hat{f}_t = f_t \circ \omega_t + (\tilde{f}_t - \omega_t) \tag{22}$$

$$\hat{i}_t = i_t \circ \omega_t + (\tilde{i}_t - \omega_t). \tag{23}$$

As mentioned in Section B.1, this model modifies the standard forget/input gates in two main ways, namely ordering the gates via the cumax activation, and supplying an auxiliary set of gates controlling fine-grained behavior. Both of these are important novelties and together allow recurrent models to better capture tree structures.

However, the UGI and refine gate can be viewed as improvements over each of these, respectively, demonstrated both theoretically (below) and empirically (Sections 3 and E.3), even on tasks involving hierarchical sequences.

**Ordered gates** Despite having the same parameter count and asymptotic efficiency as standard sigmoid gates, cumax gates seem noticeably slower and less stable in practice for large hidden sizes. Additionally, using auxiliary master gates creates additional parameters compared to the basic LSTM. Shen et al. (2018) alleviated both of these problems by defining a *downsize* operation, whereby neurons are grouped in chunks of size $C$, each of which share the same master gate values. However, this also creates an additional hyperparameter.

The speed and stability issues can be fixed by just using the sigmoid non-linearity instead of cumax. To recover the most important properties of the cumax—activations at multiple timescales—the equivalent sigmoid gate can be initialized so as to match the distribution of cumax gates at initialization. This is just uniform gate initialization (equation (9)).

However, we believe that the cumax activation is still valuable in many situations if speed and instability are not issues. These include when the hidden size is small, when extremal gate activations are desired, or when ordering needs to be strictly enforced to induce explicit hierarchical structure. For example, Section (3.1) shows that they can solve hard memory tasks by themselves.

**Master gates** We observe that the magnitudes of master gates are suboptimally normalized. A nice interpretation of gated recurrent models shows that they are a discretization of a continuous differential equation. This leads to the leaky RNN model $h_{t+1} = (1-\alpha)h_t + \alpha u_t$, where $u_t$ is the update to the model such as $\tanh(W_x x_t + W_h h_t + b)$. Learning $\alpha$ as a function of the current time step leads to the

simplest gated recurrent model[4]

$$f_t = \sigma(\mathcal{L}_f(x_t, h_{t-1}))$$
$$u_t = \tanh(\mathcal{L}_u(x_t, h_{t-1}))$$
$$h_t = f_t h_{t-1} + (1 - f_t)u_t.$$

Tallec & Ollivier (2018) show that this exactly corresponds to the discretization of a differential equation that is invariant to *time warpings* and time rescalings. In the context of the LSTM, this interpretation requires the values of the forget and input gates to be tied so that $f_t + i_t = 1$. This weight-tying is often enforced, for example in the most popular LSTM variant, the GRU (Cho et al., 2014), or our UR-gates. In a large-scale LSTM architecture search, it was found that removing the input gate was not significantly detrimental (Greff et al., 2016).

However, the ON-LSTM does not satisfy this conventional wisdom that the input and forget gates should sum to close to 1.

**Proposition 3.** *At initialization, the expected value of the average effective forget gate activation $\hat{f}_t$ is* $5/6$.

Let us consider the sum of the effective forget and input gates at initialization. Adding equations (22) and (23) yields

$$\hat{f}_t + \hat{i}_t = (f_t + i_t) \circ \omega_t + (\tilde{f}_t + \tilde{i}_t - 2\omega_t)$$
$$= \tilde{f}_t + \tilde{i}_t + (f_t + i_t - 2) \circ \omega_t.$$

Note that the master gates (19), (20) sum 1 in expectation at initialization, as do the original forget and input gates. Looking at individual units in the ordered master gates, we have $\mathbb{E}\hat{f}^{(j)} = \frac{j}{n}, \mathbb{E}\hat{i}^{(j)} = 1 - \frac{j}{n}$. Thus the above simplifies to

$$\mathbb{E}[\hat{f}_t + \hat{i}_t] = 1 - \mathbb{E}\omega_t$$

$$\mathbb{E}[\hat{f}_t^{(j)} + \hat{i}_t^{(j)}] = 1 - \frac{j}{n}(1 - \frac{j}{n})$$

$$\mathbb{E}\left[\mathbb{E}_{j \in [n]} \hat{f}_t^{(j)} + \hat{i}_t^{(j)}\right] \approx 1 - \int_0^1 x \, dx + \int_0^1 x^2 \, dx$$

$$= \frac{5}{6}.$$

The gate normalization can be fixed by re-scaling equations (22) and (23). It turns out that tying the master gates and re-scaling is exactly equivalent to the mechanism of a refine gate. In this equivalence, the role of the master and forget gates of the ON-LSTM are played by our forget and refine gate respectively.

**Proposition 4.** *Suppose the master gates $\tilde{f}_t, \tilde{i}_t$ are tied and the equations* (22)-(23) *defining the effective gates $\hat{f}_t, \hat{i}_t$ are rescaled such as to ensure $\mathbb{E}[\hat{f}_t + \hat{i}_t] = 1$ at initialization. The resulting gate mechanism is exactly equivalent to that of the refine gate.*

Consider the following set of equations where the master gates are tied ($\tilde{f}_t + \tilde{i}_t = 1, f_t + i_t = 1$) and (22)-(23) are modified with an extra coefficient (rescaling in bold):

$$\tilde{i}_t = 1 - \tilde{f}_t \tag{24}$$

$$\omega_t = \tilde{f}_t \cdot \tilde{i}_t \tag{25}$$

$$\hat{f}_t = \mathbf{2} \cdot f_t \cdot \omega_t + (\tilde{f}_t - \omega_t) \tag{26}$$

$$\hat{i}_t = \mathbf{2} \cdot i_t \cdot \omega_t + (\tilde{i}_t - \omega_t) \tag{27}$$

Now we have

$$\hat{f}_t + \hat{i}_t = \tilde{f}_t + \tilde{i}_t + 2(f_t + i_t - 1) \cdot \omega_t$$
$$= 1 + 2(f_t + i_t - 1) \cdot \omega_t$$

---

[4]In the literature, this is called the JANET (van der Westhuizen & Lasenby, 2018), which is also equivalent to the GRU without a reset gate (Chung et al., 2014), or a recurrent highway network with depth $L = 1$ (Zilly et al., 2017).

which has the correct scaling, i.e. $\mathbb{E}[\hat{f}_t + \hat{i}_t] = 1$ at initialization assuming that $\mathbb{E}[f_t + i_t] = 1$ at initialization.

But (26) can be rewritten as follows:

$$
\begin{aligned}
\hat{f} &= 2 \cdot f \cdot \omega + (\tilde{f} - \omega) \\
&= 2 \cdot f \cdot \tilde{f} \cdot (1 - \tilde{f}) + (\tilde{f} - \tilde{f} \cdot (1 - \tilde{f})) \\
&= 2f \cdot \tilde{f} - 2f \cdot \tilde{f}^2 + \tilde{f}^2 \\
&= f \cdot 2\tilde{f} - f \cdot \tilde{f}^2 - f \cdot \tilde{f}^2 + \tilde{f}^2 \\
&= f \cdot (1 - (1 - \tilde{f}))^2 + (1 - f) \cdot \tilde{f}^2.
\end{aligned}
$$

This is equivalent to the refine gate, where the master gate plays the role of the forget gate and the forget gate plays the role of the refine gate. It can be shown that in this case, the effective input gate $\hat{i}_t$ (27) is also defined through a refine gate mechanism, where $\tilde{i}_t = 1 - \tilde{f}_t$ is refined by $i_t$:

$$
\hat{i} = i \cdot (1 - (1 - \tilde{i}))^2 + (1 - i) \cdot \tilde{i}^2.
$$

Based on our experimental findings, in general we would recommend the refine gate in place of the master gate.

## B.5 GATE ABLATION DETAILS

For clarity, we formally define the gate ablations considered which mix and match different gate components.

We remark that other combinations are possible, for example combining CI with either auxiliary gate type, which would lead to CR- or CM- gates. Alternatively, the master or refine gates could be defined using different activation and initialization strategies. We chose not to consider these methods due to lack of interpretation and theoretical soundness.

**O-** This ablation uses the $\mathrm{cumax}$ activation to order the forget/input gates and has no auxiliary gates.

$$
f_t = \mathrm{cumax}(\mathcal{L}_f(x_t, h_{t-1})) \tag{28}
$$
$$
i_t = 1 - \mathrm{cumax}(\mathcal{L}_i(x_t, h_{t-1})). \tag{29}
$$

We note that one difficulty with this in practice is the reliance on the expensive $\mathrm{cumax}$, and hypothesize that this is perhaps the ON-LSTM's original motivation for the second set of gates combined with downsizing.

**UM-** This variant of the ON-LSTM ablates the $\mathrm{cumax}$ operation on the master gates, replacing it with a sigmoid activation initialized with UGI. Equations (19), (20) are replaced with

$$
u = \mathcal{U}(0,1) \tag{30}
$$
$$
b_f = \sigma^{-1}(u) \tag{31}
$$
$$
\tilde{f}_t = \sigma(\mathcal{L}_{\tilde{f}}(x_t, h_{t-1}) + b_f) \tag{32}
$$
$$
\tilde{i}_t = \sigma(\mathcal{L}_{\tilde{i}}(x_t, h_{t-1}) - b_f) \tag{33}
$$

Equations (21)-(23) are then used to define effective gates $\hat{f}_t, \hat{i}_t$ which are used in the gated update (1) or (5).

**OR-** This ablation combines ordered main gates with an auxilliary refine gate.

$$
\tilde{f}_t = \mathrm{cumax}(\mathcal{L}_{\tilde{f}}(x_t, h_{t-1}) + b_f) \tag{34}
$$
$$
r_t = \sigma(\mathcal{L}_r(x_t, h_{t-1}) + b_r) \tag{35}
$$
$$
g_t = r_t \cdot (1 - (1 - f_t)^2) + (1 - r_t) \cdot f_t^2 \tag{36}
$$
$$
i_t = 1 - g_t \tag{37}
$$

$g_t, i_t$ are used as the effective forget and input gates.

## C  ANALYSIS DETAILS

The gradient analysis in Figure 2 was constructed as follows. Let $f, r, g$ be the forget, refine, and effective gates

$$g = 2rf + (1-2r)f^2.$$

Then

$$\nabla_x g = 2rf(1-f) + (1-2r)(2f)(f(1-f)) = 2f(1-f)[r + (1-2r)f]$$
$$\nabla_y g = 2fr(1-r) + (-2f^2)r(1-r) = 2fr(1-r)(1-f)$$
$$\|\nabla g\|^2 = [2f(1-f)]^2 \big[(r+f-2fr)^2 + r^2(1-r)^2\big].$$

Substituting the relation

$$r = \frac{g-f^2}{2f(1-f)},$$

this reduces to

$$\|\nabla g\|^2 = [2f(1-f)]^2 \left[ \left( \frac{g-f^2 + 2f^2(1-f) - 2f(g-f^2)}{2f(1-f)} \right)^2 + \frac{(g-f^2)^2(2f(1-f) - g + f^2)^2}{(2f(1-f))^4} \right]$$

$$= ((g-f^2)(1-2f) + 2f^2(1-f))^2 + (g-f^2)^2 \left( 1 - \frac{g-f^2}{2f(1-f)} \right)^2.$$

Given the constraint $f^2 \le g \le 1 - (1-f)^2$, this function can be minimized and maximized in terms of $g$ to produce the upper and lower bounds in Figure 2d. This was performed numerically.

## D  EXPERIMENTAL DETAILS

To normalize the number of parameters used for models using master gates, i.e. the OM- and UM-gating mechanisms, we used a downsize factor on the main gates (see Section B.4). This was set to $C = 16$ for the synthetic and image classification tasks, and $C = 32$ for the language modeling and program execution tasks which used larger hidden sizes.

### D.1  SYNTHETIC TASKS

All models consisted of single layer LSTMs with 256 hidden units, trained with the Adam optimizer Kingma & Ba (2014) with learning rate 1e-3. Gradients were clipped at $1.0$.

The training data consisted of randomly generated sequences for every minibatch rather than iterating through a fixed dataset. Each method ran 3 seeds, with the same training data for every method.

Our version of the Copy task is a very minor variant of other versions reported in the literature, with the main difference being that the loss is considered only over the last 10 output tokens which need to be memorized. This normalizes the loss so that losses approaching 0 indicate true progress. In contrast, this task is usually defined with the model being required to output a dummy token at the first $N+10$ steps, meaning it can be hard to evaluate performance since low average losses simply indicate that the model learns to output the dummy token.

For Figure 3, the log loss curves show the median of 3 seeds, and the error bars indicate 60% confidence.

For Figure 4, each histogram represents the distribution of forget gate values of the hidden units (of which there are 256). The values are created by averaging units over time and samples, i.e., reducing a minibatch of forget gate activations of shape (`batch size, sequence length, hidden size`) over the first two diensions, to produce the average activation value for every unit.

### D.2  IMAGE CLASSIFICATION

All models used a single hidden layer recurrent network (LSTM or GRU). Inputs $x$ to the model were given in batches as a sequence of shape (`sequence length, num channels`), (e.g. (1024,3)

for CIFAR-10), by flattening the input image left-to-right, top-to-bottom. The outputs of the model of shape `(sequence length, hidden size)` were processed independently with a single ReLU hidden layer of size 256 before the final fully-connected layer outputting softmax logits. All training was performed with the Adam optimizer, batch size 50, and gradients clipped at 1.0. MNIST trained for 150 epochs, CIFAR-10 used 100 epochs over the training set.

**Table 2**    All models (LSTM and GRU) used hidden state size 512. Learning rate swept in $\{2e-4, 5e-4, 1e-3, 2e-3\}$ with three seeds each.

Table 2 reports the highest validation score found. The GRU model swept over learning rates $\{2e-4, 5e-4\}$; all methods were unstable at higher learning rates.

Figure 5 shows the median validation accuracy with quartiles (25/75% confidence intervals) over the seeds, for the best-performing stable learning rate (i.e. the one with highest average validation score on the final epoch).

**Table 3**    The UR-LSTM and UR-GRU used 1024 hidden units for the sequential and permuted MNIST task, and 2048 hidden units for the sequential CIFAR task. The vanilla LSTM baseline used 512 hidden units for MNIST and 1024 for CIFAR. Larger hidden sizes were found to be unstable.

Zoneout parameters were fixed to reasonable default settings based on Krueger et al. (2016), which are $z_c = 0.5, z_h = 0.05$ for LSTM and $z = 0.1$ for GRU. When zoneout was used, standard Dropout (Srivastava et al., 2014) with probability 0.5 was also applied to the output classification hidden layer.

### D.3    LANGUAGE MODELING

Hyperparameters are taken from Rae et al. (2018) tuned for the vanilla LSTM, which consist of (chosen parameter bolded out of sweep): $\{1,2\}$ LSTM layer, $\{0.0, 0.1, 0.2, \mathbf{0.3}\}$ embedding dropout, $\{yes, \mathbf{no}\}$ layer norm, and $\{\mathbf{shared}, \text{not shared}\}$ input/output embedding parameters. Our only divergence is using a hidden size of 3072 instead of 2048, which we found improved the performance of the vanilla LSTM. Training was performed with Adam at learning rate 1e-3, gradients clipped to 0.1, sequence length 128, and batch size 128 on TPU. The LSTM state was reset between article boundaries. Figure 6b shows smoothed validation perplexity curves showing the 95% confidence intervals over the last 1% of data.

**Reinforcement Learning**    The Active Match and Passive Match tasks were borrowed from Hung et al. (2018) with the same settings. For Figures 7 and 10, the discount factor in the environment was set to $\gamma = .96$. For Figure 8, the discount factor was $\gamma = .998$. Figure 10 corresponds to the full Active Match task in Hung et al. (2018), while Figure 8 is their version with small distractor rewards where the apples in the distractor phase give 1 instead of 5 reward.

### D.4    PROGRAM EVALUATION

Protocol was taken from Santoro et al. (2018) with minor changes to the hyperparameter search. All models were trained with the Adam optimizer, the *Mix* curriculum strategy from Zaremba & Sutskever (2014), and batch size 128.

RMC: The RMC models used a fixed memory slot size of 512 and swept over $\{2,4\}$ memories and $\{2,4\}$ attention heads for a total memory size of 1024 or 2048. They were trained for 2e5 iterations.

LSTM: Instead of two-layer LSTMs with sweeps over skip connections and output concatenation, single-layer LSTMs of size 1024 or 2048 were used. Learning rate was swept in $\{5e-4, 1e-3\}$, and models were trained for 5e5 iterations. Note that training was still faster than the RMC models despite the greater number of iterations.

### D.5    ADDITIONAL DETAILS

**Implementation Details**    The inverse sigmoid function (9) can be unstable if the input is too close to $\{0,1\}$. Uniform gate initialization was instead implemented by sampling from the distribution $\mathcal{U}[1/d, 1-1/d]$ instead of $\mathcal{U}[0,1]$, where $d$ is the hidden size, to avoid any potential numerical edge cases. This choice is justified by the fact that with perfect uniform sampling, the expected smallest and largest samples would be $1/(d+1)$ and $1-1/(d+1)$.

For distributional initialization strategies, a trainable bias vector was sampled independently from the chosen distribution (i.e. equation (17) or (9)) and added/subtracted to the forget and input gate ((2)-(3)) before the non-linearity. Additionally, each linear model such as $W_{xf}x_t + W_{hf}h_{t-1}$ had its own trainable bias vector, effectively doubling the learning rate on the pre-activation bias terms on the forget and input gates. This was an artifact of implementation and not intended to affect performance.

The refine gate update equation (12) can instead be implemented as

$$g_t = r_t \cdot (1 - (1 - f_t)^2) + (1 - r_t) \cdot f_t^2$$
$$= 2r_t \cdot f_t + (1 - 2r_t) \cdot f_t^2$$

**Permuted image classification** In an effort to standardize the permutation used in the Permuted MNIST benchmark, we use a particular deterministic permutation rather than a random one. After flattening the input image into a one-dimensional sequence, we apply the *bit reversal* permutation. This permutation sends the index $i$ to the index $j$ such that $j$'s binary representation is the reverse of $i$'s binary representation. The intuition is that if two indices $i, i'$ are close, they must differ in their lower-order bits. Then the bit-reversed indices will be far apart. Therefore the bit-reversal permutation destroys spatial and temporal locality, which is desirable for these sequence classification tasks meant to test long-range dependencies rather than local structure.

```python
def bitreversal_po2(n):
    m = int(math.log(n) / math.log(2))
    perm = np.arange(n).reshape(n, 1)
    for i in range(m):
        n1 = perm.shape[0] // 2
        perm = np.hstack((perm[:n1], perm[n1:]))
    return perm.squeeze(0)

def bitreversal_permutation(n):
    m = int(math.ceil(math.log(n) / math.log(2)))
    N = 1 << m
    perm = bitreversal_po2(N)
    return np.extract(perm < n, perm)
```

# E ADDITIONAL EXPERIMENTS

## E.1 SYNTHETIC FORGETTING

Figure 4 on the Copy task demonstrates that extremal gate activations are necessary to solve the task, and initializing the activations near $1.0$ is helpful.

This raises the question: what happens if the initialization distribution does not match the task at hand; could the gates learn back to a more moderate regime? We point out that such a phenomenon could occur non-pathologically on more complex setups, such as a scenario where a model trains to remember on a Copy-like task and then needs to "unlearn" as part of a meta-learning or continual learning setup.

Here, we consider such a synthetic scenario and experimentally show that the addition of a refine gate helps models train much faster while in a saturated regime with extremal activations. We also point to the poor performance of C- outside of synthetic memory tasks when using our high hyperparameter-free initialization as more evidence that it is very difficult for standard gates to unlearn undesired saturated behavior.

For this experiment, we initialize the biases of the gates extremely high (effective forget activation $\approx \sigma(6)$. We then consider the Adding task (Section 3.1 of length 500, hidden size 64, learning rate 1e-4. The R-LSTM is able to solve the task, while the LSTM is stuck after 1e4 iterations.

## E.2 REINFORCEMENT LEARNING

Figures 7 and 8 evaluated our gating methods with the LSTM and RMA models on the Passive Match and Active Match tasks, with and without distractors. We additionally ran the agents on an even harder

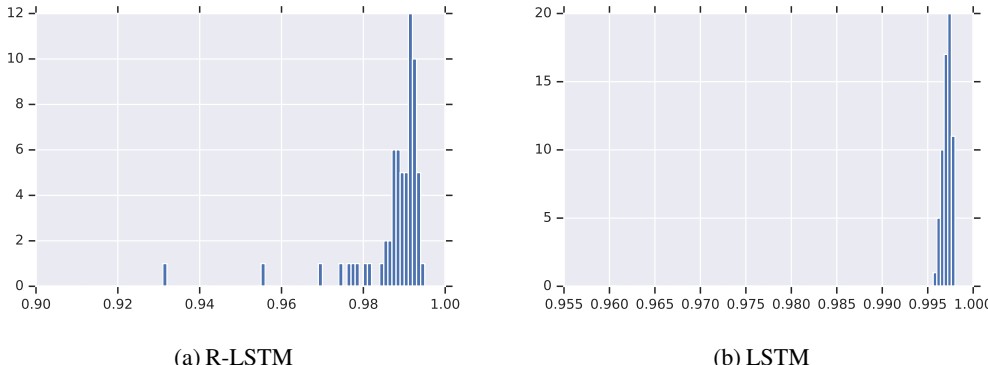

(a) R-LSTM             (b) LSTM

Figure 9: Distribution of forget gate activations after extremal initialization, and training on the Adding task. The UR-LSTM is able to learn much faster in this saturated gate regime while the LSTM does not solve the task. The smallest forget unit for the UR-LSTM after training has characteristic timescale over an order of magnitude smaller than that of the LSTM.

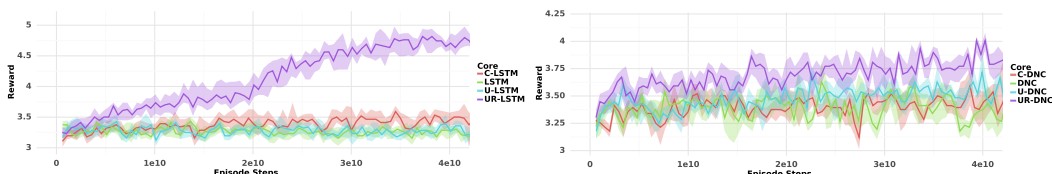

Figure 10: The full Active Match task with large distractor rewards, using agents with LSTM or DNC recurrent cores.

version of the Active Match task with larger distractor rewards (the full Active Match from Hung et al. (2018)). Learning curves are shown in Figure 10. Similarly to the other results, the UR- gated core is noticeably better than the others. For the DNC model, it is the only one that performs better than random chance.

### E.3  PROGRAM EXECUTION

The Learning to Execute (Zaremba & Sutskever, 2014) dataset consists of algorithmic snippets from a programming language of pseudo-code. An input is a program from this language presented one character at a time, and the target output is a numeric sequence of characters representing the execution output of the program. There are three categories of tasks: Addition, Control, and Program, with distinctive types of input programs. We use the most difficult setting from Zaremba & Sutskever (2014), which uses the parameters `nesting=4, length=9`, referring to the nesting depth of control structure and base length of numeric literals, respectively. Examples of input programs are shown in previous works (Zaremba & Sutskever, 2014; Santoro et al., 2018).

We are interested in this task for several reasons. First, we are interested in comparing against the C- and OM- gate methods, because

- The maximum sequence length is fairly long (several hundred tokens), meaning our $T_{max}$ heuristic for C- gates is within the right order of magnitude of dependency lengths.
- The task has highly variable sequence lengths, wherein the standard training procedure randomly samples inputs of varying lengths (called the "Mix" curriculum in Zaremba & Sutskever (2014)). Additionally, the Control and Program tasks contain complex control flow and nested structure. They are thus a measure of a sequence model's ability to model dependencies of differing lengths, as well as hierarchical information. Thus we are interested in comparing the effects of UGI methods, as well as the full OM- gates which are designed for hierarchical structures (Shen et al., 2018).

Finally, this task has prior work using a different type of recurrent core, the Relational Memory Core (RMC), that we also use as a baseline to evaluate our gates on different models Santoro et al. (2018). Both the LSTM and RMC were found to outperform other recurrent baselines such as the Differential Neural Computer (DNC) and EntNet.

Training curves are shown in Figure 11, which plots the median accuracy with confidence intervals. We point out a few observations. First, despite having a $T_{max}$ value on the right order of magnitude, the C-gated methods have very poor performance across the board, reaffirming the chrono initialization's high sensitivity to this hyperparameter.

Second, the U-LSTM and U-RMC are the best methods on the Addition task. Additionally, the UR-RMC vs. RMC on Addition is one of the very few tasks we have found where a generic substitution of the UR- gate does not improve on the basic gate. We have not investigated what property of this task caused these phenomena.

Aside from the U-LSTM on addition, the UR-LSTM outperforms all other LSTM cores. The UR-RMC is also the best core on both Control and Program, the tasks involving hierarchical inputs and longer dependencies. For the most part, the improved mechanisms of the UR- gates seem to transfer to this recurrent core as well. We highlight that this is not true of similar gating mechanisms. In particular, the OM-LSTM, which is supposed to model hierarchies, has good performance on Control and Program as expected (although not better than the UR-LSTM). However, the OM- gates' performance plummets when transferred to the RMC core.

Interestingly, the -LSTM cores are consistently better than the -RMC versions, contrary to previous findings on easier versions of this task using similar protocol and hyperparameters Santoro et al. (2018). We did not explore different hyperparameter regimes on this more difficult setting.

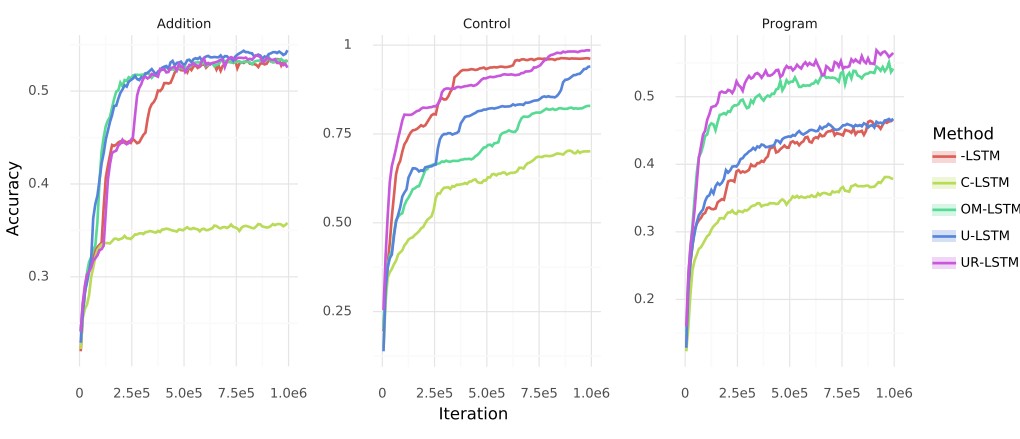

(a) LSTM - Learning to Execute (nesting=4, length=9)

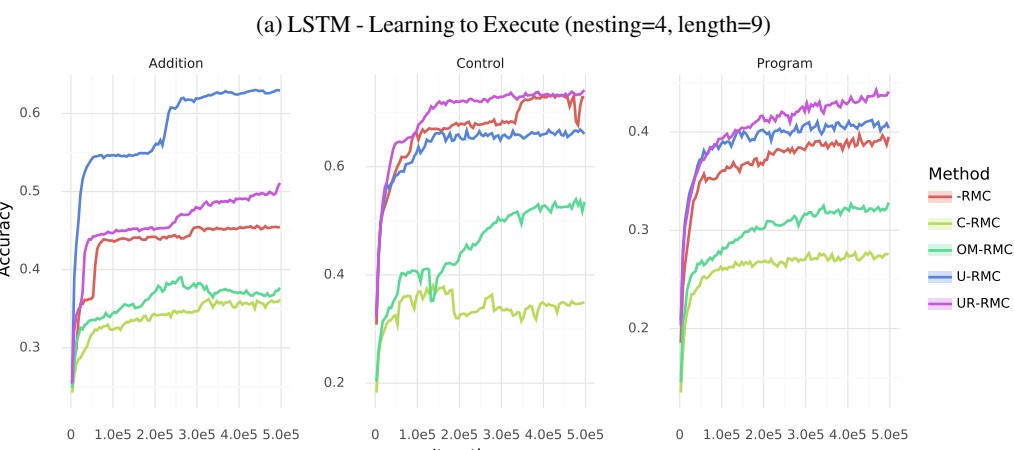

(b) RMC - Learning to Execute (nesting=4, length=9)

Figure 11: Program Execution evaluation accuracies.

