# OpenReview forum: "Improving the Gating Mechanism of Recurrent Neural Networks"
_ICLR.cc/2020/Conference — Reject_

### Official Review · AnonReviewer2 · 2019-10-22
**Official Blind Review #2**

**Rating:** 3

**Review:**

This paper introduces two novel techniques to help long term signal propagation in RNNs. One is an initialization strategy which uses inverse sigmoid function to avoid the decay of the contribution of the input earlier in time and another is a new design of a refine gate which pushes the value of the gate closer to 0 or 1.  The authors conduct exhaustive ablation and empirical studies on copy task, sequential MNIST, language modeling and reinforcement learning.

Though the experiment section is solid, I still vote for rejection for the following reasons:

1. The writing in the description of the UGI and refine fate is not clear.
a. The authors compares UGI to standard initialization but where is the standard initialization? I do not see "standard initialization" clearly defined in the paper.
b. I am not convinced how the UGI gate help avoid decaying of the input. There is a proposition 2 trying to explain some part of the mechanism of UGI. But the proposition is never proved anywhere and I am not sure why this proposition is important. More explanations are needed. Also this proposition is far away from the place the authors introduce the UGI. The authors may want to refer it in the place introducing UGI.
c. Similar to proposition 2, proposition 3 is not explained and proved in the paper. It is hard for me to analyze the importance of these two propositions. Overall, propositions 2 and 3 look isolated in the section.
d. Proposition 1 looks like a definition. Not sure why the authors name it as a proposition.

2. Even though the title of the paper is "improving the gating mechanism of recurrent neural networks", the authors try to solve signal propagation problems. It is unclear why "gate" is important. Maybe other designs of the recurrent neural network can satisfy better the desiderata the authors want. Based on my limited knowledge, the initialization the authors mention (saturation) is exactly from the need of using a sigmoid gate. The importance of using "gate" should be discussed.

3. The authors shrink the space before and after headings in the paper. I think this is not allowed in ICLR. It would be better that the authors correct the spacing in the revised version.


Minors:
1. page 1 second paragraph: repeated “proven”
2. page 1 second last paragraph: “due” -> “due to”
3. page 2 second last paragraph: repeated “be”
4. page 2 Equation (4) and (5): using some symbols like \odot for element wise multiplication will be good for the readers.




**Experience Assessment:**

I have read many papers in this area.

**Review Assessment: Checking Correctness Of Derivations And Theory:**

I assessed the sensibility of the derivations and theory.

**Review Assessment: Checking Correctness Of Experiments:**

I assessed the sensibility of the experiments.

**Review Assessment: Thoroughness In Paper Reading:**

I read the paper at least twice and used my best judgement in assessing the paper.

---

> ### Author Response · Authors · 2019-11-08
> **Thank you for your feedback**
>
> We thank the reviewer for the helpful feedback and suggestions, which we respond to below.
>
> >>> The writing in the description of the UGI and refine fate is not clear.
>
> We have improved the motivation and description of UGI and the refine gate in the revision. Further feedback on how they can be further clarified is appreciated.
>
> >>> The authors compares UGI to standard initialization but where is the standard initialization?
>
> Standard initialization was stated to be initializing the bias term to a constant; for example linear models usually initialize the bias to 0 (Section 2.2). In our experiments, the vanilla LSTM used 1.0 bias initialization, which has become the standard for LSTMs (Gers et al.). This detail has been clarified in Section 2.2 and 3.
>
> >>> I am not convinced how the UGI gate help avoid decaying of the input
>
> Please see our shared response to all reviewers.
>
>
> >>> On propositions
>
> The central principle of gated RNNs is that the values of gate activations control the timescales that the model can address. Propositions 2 and 3 formalize how our gate modifications affect the timescales of the recurrent model. However, we agree that Propositions 2 and 3 are somewhat tangential to the main points of the paper and have moved them to Appendix B.2.
>
>
> >>> Even though the title of the paper is "improving the gating mechanism of recurrent neural networks", the authors try to solve signal propagation problems. It is unclear why "gate" is important.
>
> The above principle means that the performance of gated RNNs (which are the dominant form of RNN, for reasons outlined in Section 1 and 2) is closely tied to the activations and learnability of their gates. Thus our improvements to the gates improve recurrent models at large. We hope that our revised introduction and background sections have clarified this motivation.
>
>
> >>> Minors
>
> Thanks for pointing out several mistakes which have now been fixed. In particular, we have used the suggestion to use the Hadamard multiplication symbol to emphasize the elementwise multiplication, for example in Equation (1).

---

### Official Review · AnonReviewer1 · 2019-10-23
**Official Blind Review #1**

**Rating:** 6

**Review:**


This paper introduces and studies modifications to gating mechanisms in RNNs.
The first modification is uniform gate initialization. Here the biases in the forget and input gate bias are sampled such that after the application of the sigmoid the values are in the range (1/d, 1-1/d) where d is the dimensionality of the hidden space for the bias. The second modification is the introduction of a refine gating mechanism with a view to allow for gradients to flow when the forget gates f_t in the LSTM updates are near saturation. The idea is to create an effective gate g = f_t +/- phi(f_t, x_t). The paper proposes using phi (f_t, x_t) = f_t(1-f_t) * (2*r_t-1) where (r_t is between 0 or 1). The effect of such a change is that g_t can reach values of 0.99 when the value of f_t is 0.9 allowing gradients to flow more freely through the parameters that constitute the forget gate. Overall the change corresponds to improving gradient flow for the forget gate by interpolating between f_t^2 and (1-f_t)^2. i.e. the authors note that the result of these changes is that it corresponds to sampling biases from a heavier tailed distribution while the refine gate (by allowing the forget gate to reach values close to 0 and 1), allows for capturing information on a much longer time scale.


The paper studies various combinations of the two changes proposed to gating architectures. Other baselines include a vanilla LSTM, a Chrono initialized LSTM, and an ordered Neuron LSTM. The models are trained on several synthetic and real world tasks. On the copy and add tasks, the LSTMs that contain the refine gate converge the fastest. A similar story is observed on the task of pixel by pixel image classification. The refine gate was also adapted to memory architectures such as the DNC and RMA where it was found to improve performance on two different tasks.

Overall, the paper is written well, I like the (second) idea of the refine gate and the contributions are explained in an accessible manner. While I'm not entirely convinced about the proposed initialization scheme but across the many different tasks tried, the use of the refine gate does appear to give performance improvements that lead me to conclude that this aspect of the work is a solid contribution to the literature.

Questions and comments:
* This manuscript already quite long and has several formatting issues. Several of the figures are unreadable when printed. For example, every piece of text on Figure 2(d) is unreadable on paper. Figure 3 and 5 are difficult to read; they contain too many alternatives with a colour scheme that makes it difficult to distinguish between them -- consider displaying a subset of the options via a plot and using a table to display (# steps to convergence) as a metric instead. It also appears as if the caption for Table 6 is deleted?
* I think that for this approach to work, two conditions need to be satisfied (a) there must be foreseeable improvements in the use of a forget gate that can reach values close to 0/1 for the task at hand and (b) r_t needs to function well despite not being too close to 0 or 1 (lest its parameters suffer from gradient flow issues).
  * Was there any visualizations done on whether (a) happened? i.e. for the URLSTMs that performed well, were the values of the forget gate closer to 0/1 than the baselines?
  * What were typical values of r_t, did the models need the refine gate to reach values close to 0 or 1 for the overall approach to work?

**Experience Assessment:**

I have read many papers in this area.

**Review Assessment: Checking Correctness Of Derivations And Theory:**

I assessed the sensibility of the derivations and theory.

**Review Assessment: Checking Correctness Of Experiments:**

I assessed the sensibility of the experiments.

**Review Assessment: Thoroughness In Paper Reading:**

I read the paper thoroughly.

---

> ### Author Response · Authors · 2019-11-08
> **Thank you for your feedback**
>
> We appreciate the reviewer’s detailed reading of the paper and thoughtful comments, suggestions, and questions. We have responded to these below.
>
>
> >>> This manuscript already quite long and has several formatting issues...
>
> We agree with the suggestions and have improved the labels and captions of many figures. We have updated Figure 3 on a log scale which helps distinguish the curves. Figure 5 conveys information that a table of numbers may not be able to; for example, it shows that the UR-LSTM converges both faster and to a higher maximum than other methods, and accounts for confidence intervals. We will continue revising Figure 2 for readability.
>
>
> >>> I think that for this approach to work, two conditions need to be satisfied (a) there must be foreseeable improvements in the use of a forget gate that can reach values close to 0/1 for the task at hand and (b) r_t needs to function well despite not being too close to 0 or 1 (lest its parameters suffer from gradient flow issues)
>   * Was there any visualizations done on whether (a) happened? i.e. for the URLSTMs that performed well, were the values of the forget gate closer to 0/1 than the baselines?
>
> We agree with these insights. We believe that Figure 4 and Figure 9 show clear empirical support for these principles, in particular that
>   *  How close the forget gate activations are to 1.0 affect whether the models are able to solve this memory task
>   *  The addition of a refine gate improves the ability of the model to learn more extremal activations, enabling it to solve the task
>
>
> >>> What were typical values of r_t, did the models need the refine gate to reach values close to 0 or 1 for the overall approach to work?
>
> We note that as long as one of the gates is not saturated (either the original gate or the refine gate), the model can still learn.
> We did not find any scenarios empirically in which the refine gate saturated and prevented the overall approach from working.
>
>
> >>> While I'm not entirely convinced about the proposed initialization scheme...
>
> We believe that in addition to the refine gate, UGI is also quite motivated and shows clear empirical benefits, which we summarized in the shared response to all reviewers.

---

### Official Review · AnonReviewer3 · 2019-10-24
**Official Blind Review #3**

**Rating:** 3

**Review:**

This paper proposes to improve the learnability of the gating mechanism in RNN by two modifications on the standard RNN structure, uniform gate initialization and refine gate. The authors give some propositions to show that the refine gate can maintain an effective forget effect within a larger range of timescale. The authors conduct experiments on four different tasks and compare the proposed modification with baseline methods.

Strong points:
1. The authors propose a new refine structure that seems to have a longer "memory".
2. The authors designed a good synthetic experiment to demonstrate whether the proposed refine structure can help to remember information in longer sequence.

Weak points:
1. There are several parts in the experiment that are not very convincing.
     a. In Figure 3(a), where are the other baselines? Are they performing too badly so that they can not show up in the figure? It needs more explanation.
     b.  In Figure 3(b), actually a lot of methods are performing similar, while some methods converge similarly. What is the reason?
2. It is not defined why the uniform gate initialization works.
3. The proposed results actually not always perform the best. For instance, in Table 3, purely using the UR-LSTM only achieve good results on sMNIST. What is the reason? The proposed method seems not very general.


**Experience Assessment:**

I have read many papers in this area.

**Review Assessment: Checking Correctness Of Derivations And Theory:**

I assessed the sensibility of the derivations and theory.

**Review Assessment: Checking Correctness Of Experiments:**

I carefully checked the experiments.

**Review Assessment: Thoroughness In Paper Reading:**

I read the paper at least twice and used my best judgement in assessing the paper.

---

> ### Author Response · Authors · 2019-11-08
> **Thank you for your feedback**
>
> We thank the reviewer for pointing out potentially confusing aspects of the submission, which we address below.
>
> >>>   a. In Figure 3(a), where are the other baselines? Are they performing too badly so that they can not show up in the figure? It needs more explanation.
>      b.  In Figure 3(b), actually a lot of methods are performing similar, while some methods converge similarly. What is the reason?
>
> Some baselines seem to not show up because their curves are actually overlapping. This is because the data in our experiments were controlled between methods (i.e., every method saw the same training and testing minibatches for a given seed) to reduce potential variance. Consequently, models that were unable to make any progress on this task showed identical performance. We have added these details in the caption and in Appendix [].
>
> We have additionally re-plotted these results on a log scale, which distinguishes the learning curves better.
>
>
> >>> There are several parts in the experiment that are not very convincing.
> Aside from details in Figure 3, could you point to other experiments that could be clarified?
>
>
> >>> It is not defined why the uniform gate initialization works.
> Please see our shared response to all reviewers.
>
>
> >>> The proposed results actually not always perform the best. For instance, in Table 3, purely using the UR-LSTM only achieve good results on sMNIST. What is the reason? The proposed method seems not very general.
>
> We respectfully disagree with this conclusion. In response to Table 3:
>     -  First, there was a typo that has been fixed: the “GRU + Zoneout” method that achieved SOTA on sCIFAR is actually our method “UR-GRU + Zoneout”. We apologize for the confusion this may have caused
>     -  On sCIFAR, Table 3 shows that the UR-LSTM by itself improves on a vanilla LSTM or Transformer by almost 10% accuracy. It is also the best recurrent model on pMNIST already
>     -  As mentioned in that section, our gate modifications synergize with other methods such as regularization and auxiliary losses. Table 3 shows how it synergizes with a regularization technique to achieve SOTA on sCIFAR, and we expect that the auxiliary loss of the r-LSTM (Trinh et al. 2018) can be applied to the UR-LSTM for further improvements on all datasets
>     -  We show non-recurrent models for completeness, but they are not directly comparable to ours. For example, they should be expected to be better on permuted datasets due to having global receptive field
>
> Overall, we believe that our method is very general and robust, which is shown throughout the Experiments section, where it improves on multiple types of recurrent models across many different domains.

---

### Public Comment · ~Super_User1 · 2019-10-23
**Paper modification date updated**

This paper was previously desk-rejected by mistake. It has been placed back in the submission pool.

Best,

OpenReview Team.

---

### Author Response · Authors · 2019-11-08
**Updated Paper and Response to Reviews**

We thank all the reviewers for their comments and feedback. We have uploaded a revised draft of the paper which we believe substantially improves the clarity of the submission and addresses the concerns that the reviewers raised. We highlight the most important changes below, as well as address shared feedback among the reviewers.


*Presentation and Formatting*

We apologize for the presentation issues and have fixed them to improve readability, including
    -  Increased line spacing and header separation and larger table fonts
    -  Larger and more readable figure labels
    -  Improved notation for gates


*Exposition*

    -  We have expanded on the motivation and description of our methods in Sections 1 and 2, and added a subsection (2.1) that summarizes the overall model with explicit equations for the UR-LSTM.
    -  We have added a brief discussion about the overall effectiveness of our methods (Section 3.5). In summary, we believe that the robust empirical improvement of our simple modification over the time-tested LSTM is an important contribution in itself.


*Uniform gate initialization*

We have clarified the motivation and description of UGI, which can be seen as a hyperparameter-free heuristic for improving initialization of the gates by letting neurons forget at different rates.

Empirically, we point out that UGI shows improvements on many experiments
    -  On the Copy task, the refine gate alone (R-LSTM) is stuck at baseline. Simply changing the initialization to UGI (UR-LSTM) solves the task very quickly
    -  On sequential image classification, we have added the R-LSTM ablation, which performs worse than the U-LSTM ablation (Figure 5). The isolated initialization change (e.g. from LSTM to U-LSTM, or R-LSTM to UR-LSTM) provides substantial improvements on these tasks
    -  On language modeling, the U-LSTM alone improves over the SOTA LSTM baseline and matched the more specialized ON-LSTM baseline (Figure 6)

Theoretically, we reiterate here why UGI intuitively helps
    -  The central principle of gated RNNs is that the values of gate activations control the timescales that the model can address. Section 2.2 defines the “characteristic timescale”, which implies that forget gate activations near 1.0 are necessary for long-term memory. Similar observations have been made before (Tallec et al. 2018).
    -  This phenomenon is also empirically supported in Figure 4, which shows that the methods which are able to solve the difficult memory task have more forget gate values near 1
    -  By initializing the activations uniformly throughout [0,1], UGI removes the bias hyperparameter and addresses a wider range of timescales including long-term dependencies. This explains the empirical improvements previously noted
    -  A more thorough discussion of the theoretical properties including how it affects the timescale distribution has been moved to Appendix B.2 and B.3


*Contributions*
    -  Even disregarding UGI, we emphasize that the refine gate is a novel and theoretically justified contribution that can improve any gated model
    -  We have added pseudocode snippets of the proposed methods in Appendix A, which consist of only modifying 2 lines of code each. We again highlight that these small changes translate to broad empirical improvements
    -  Overall, due to the simplicity and principled nature of these modifications, and the ubiquity of “gates” in machine learning models, we believe that our contributions can be a valuable tool for practitioners.


We have responded to other comments individually. We encourage the reviewers to look at the improved draft, and look forward to hearing further feedback on the submission.

---

### Decision · Program_Chairs · 2019-12-19

**Decision:**

Reject

**Comment:**

This submission proposes a new gating mechanism to improve gradient information propagation during back-propagation when training recurrent neural networks.

Strengths:
-The problem is interesting and important.
-The proposed method is novel.

Weaknesses:
-The justification and motivation of the UGI mechanism was not clear and/or convincing.
-The experimental validation is sometimes hard to interpret and the proposed improvements of the gating mechanism are not well-reflected in the quantitative results.
-The submission was hard to read and some images were initially illegible.

The authors improved several of the weaknesses but not to the desired level.

AC agrees with the majority recommendation to reject.